# Central Compartment Neck Dissection in Laryngeal and Hypopharyngeal Squamous Cell Carcinoma: Clinical Considerations

**DOI:** 10.3390/cancers15030804

**Published:** 2023-01-28

**Authors:** Alberto Deganello, Alessandra Ruaro, Tommaso Gualtieri, Giulia Berretti, Vittorio Rampinelli, Daniele Borsetto, Sabino Russo, Paolo Boscolo-Rizzo, Marco Ferrari, Francesco Bussu

**Affiliations:** 1Otolaryngology Head and Neck Surgery Department, IRCCS National Cancer Institute (INT), 20133 Milan, Italy; 2Section of Otolaryngology-Head and Neck Surgery Department, University of Padova, 35128 Padova, Italy; 3Otolaryngology Head and Neck Surgery Department, University of Brescia, 25123 Brescia, Italy; 4Department of ENT Surgery, Cambridge University Hospitals, Cambridge CB2 0QQ, UK; 5Otolaryngology Head and Neck Surgery Department, IRCCS National Cancer Institute (INT) “Giovanni Paolo II”, 70124 Bari, Italy; 6Department of Medical, Surgical and Health Sciences, Section of Otolaryngology, University of Trieste, 34127 Trieste, Italy; 7Otolaryngology Division, Department of Medicine, Surgery and Pharmacy, University of Sassari, 07100 Sassari, Italy

**Keywords:** central neck dissection, paratracheal lymph nodes dissection, level VI, laryngeal cancer, hypopharyngeal cancer, squamous cell carcinoma, management, prognosis, elective neck dissection

## Abstract

**Simple Summary:**

The central compartment of the neck is not cleared during a comprehensive neck dissection, which usually encompasses neck levels I to V. When involved by clinically evident lymph node metastases, central neck compartment dissection is mandatory; on the other hand, precise indications regarding the elective surgical management of occult lymph node metastases at this level are still a matter of debate. Laryngeal and hypopharyngeal squamous cell carcinomas can spread to regional lymph nodes located in the central neck compartment. This anatomic region contains the laryngeal nerves, the thyroid, and parathyroid glands, and its boundaries are represented by major vessels. This paper will focus on relevant clinical aspects that must be considered in order to formulate a judicious balance between the probability of dealing with occult disease and the possibility of being confronted with unnecessary iatrogenic risks.

**Abstract:**

Metastatic lymph node involvement represents the most relevant prognostic factor in head and neck squamous cell carcinomas (HNSCCs), invariably affecting overall survival, disease-specific survival, and relapse-free survival. Among HNSCCs, laryngeal and hypopharyngeal cancers are known to be at highest risk to metastasize to the central neck compartment (CNC). However, prevalence and prognostic implications related to the CNC involvement are not well defined yet, and controversies still exist regarding the occult metastasis rate. Guidelines for the management of CNC in laryngeal and hypopharyngeal cancers are vague, resulting in highly variable selection criteria for the central neck dissection among different surgeons and institutions. With this review, the authors intend to reappraise the existing data related to the involvement of CNC in laryngeal and hypopharyngeal malignancies, in the attempt to define the principles of management while highlighting the debated aspects that are lacking in evidence and consensus. Furthermore, as definition and boundaries of the CNC have changed over the years, an up-to-date anatomical–surgical description of the CNC is provided.

## 1. Introduction

Cervical metastatic lymph nodal disease represents the most relevant prognostic factor in head and neck squamous cell carcinoma (HNSCC), strongly affecting overall survival, disease specific survival, and relapse-free survival [1]. HNSCCs first metastasize through lymphatic vessels, rather than spreading hematogeneously, and the overall rate of nodal metastasis reported by several studies ranges between 20% and 50% [2,3].

A correlation between the tumor primary site and the distribution of lymphatic metastases in the neck has been demonstrated, outlining the presence of preferential anatomical lymphatic pathways. Moreover, a predictable pattern of neck metastasis distribution has been identified, for each primary site, by some large retrospective series [1,4,5,6] that investigated this correlation. However, in these series, the metastatic distribution to the central neck compartment (CNC) was not reported.

Laryngeal and hypopharyngeal HNSCCs are known to be prone to metastasize to the CNC [7,8], and these malignancies are nowadays considered the primary sites at highest risk for this peculiar regional spread. Nevertheless, controversies exist about the prevalence of CNC involvement, and the true incidence of microscopic disease is still unknown [9]. In a clinically positive neck, the surgical standard of care is a therapeutic neck dissection; instead, the management of a clinically negative neck is still a debated issue that follows a probabilistic approach. Therefore, an elective neck dissection or elective neck irradiation is recommended when the estimated risk of microscopic disease exceeds 15–20% [10,11,12], otherwise a watchful waiting policy is preferred. In laryngeal and hypopharyngeal cancers, guidelines regarding elective CNC dissection have not been well defined. Notably, the major concern against surgical elective treatment is represented by morbidities associated to paratracheal/recurrential lymph node (PTLN) dissection.

The lack of specific guidelines resulted in highly different and inconsistent approaches to CNC among institutions. Therefore, clarifying the prevalence and the prognostic implications of CNC involvement is critical to establish an appropriate management for the treatment of laryngeal and hypopharyngeal SCCs.

As definition and boundaries of the CNC has changed over the years, an up-to-date anatomical–surgical description of the CNC is herein provided. Furthermore, the existing data about the prevalence of CNC involvement in laryngeal and hypopharyngeal SCC, the rate of occult metastasis, and the correlation between the lateral and CNC involvement are critically analyzed. Secondly, this narrative review intends to investigate the prognostic role of CNC involvement. The most relevant evidence will be discussed in the attempt to delineate possible indications for the management of the CNC.

## 2. Anatomical Considerations on the Central Neck Compartment

The human body contains approximately 500 lymph nodes, 200 of which belong to the head and neck region. In the late 1930s, the lymphatic system of the neck was seminally described by Rouvière [13], and in the early 1990s a topographical subdivision of neck lymph nodes was introduced, internationally accepted [14], and successively updated over the years up to 2008 [15,16]. The classification currently includes seven lymph node levels, dividing the neck in a lateral compartment (level I to V), and in a central or anterior compartment (level VI and VII), whereas lymph nodes sited in the region not included within the seven levels are nominated by the specific anatomical region they belong (e.g., retropharyngeal, periparotid nodes).

Level VI boundaries are defined superiorly by the hyoid bone, inferiorly by the sternum notch, and laterally by the medial margin of carotid arteries [15], as the sternohyoid muscle, considered as the traditional surgical lateral landmark, is not easily definable at imaging [17]. Anatomically, radiographically, and surgically, three lymph node groups have been identified within the central compartment of the neck [18,19]. The first group is represented by prelaryngeal lymph nodes, which remain detectable in only 50% of the adults. In most adult patients, there is usually a single, fairly constant lymph node, the Delphian lymph node, at the midline, though there could be two, three, or four nodes located together [20,21]. The second group are the pretracheal lymphnodes, which account for six to eight nodes within the fibro-fatty tissue anterior to the trachea. These are disposed in the area between the thyroid isthmus and the thymus, if present, or the left subclavian vein [20], thus caudally extending beyond the boundaries of the level VI. The paratracheal lymphnodes (PTLNs, recurrent chain) consist of 3 to 30 nodes that are situated along the tracheoesophageal groove, becoming more dispersed towards the superior end of the recurrent laryngeal nerve. Cadaveric dissections performed by Harrison demonstrated the presence of 2 up to 10 PTLN on each side, with only one or two nodes located above the suprasternal notch [22]. The total number of PTLNs on the right side tends to be higher than the contralateral side, probably due to the wider angle between the right inferior laryngeal nerve and the carotid artery [19].

It is evident how the conventional inferior limit of level VI, the suprasternal notch, highly differs among patients since it is dependent on the anatomic conformation of the thoracic inlet. PTLNs and the pretracheal nodes extend below this level, taking part to the superior mediastinal nodes [16,23]. Martins himself stated that, ‘no attempt was made to differentiate PTLN (level VI) from other mediastinal nodes’ [24]. Therefore, level VI, in particular in its inferior limit, does not contain the whole group of nodes at risk of central nodal metastasis, confirming that the suprasternal notch is not an appropriate landmark for the dissection [8].

Level VII represents the inferior extension of the paratracheal nodes (Figure 1). It encompasses nodes located within the area defined by the suprasternal notch above, the innominate artery, and left brachiocephalic vein below that should therefore represent the landmarks of the inferior extent of the dissection. Mediastinal lymph nodes located below these vessels require a thoracic access, and for this reason they were not included in level VII [16].

In 1997, lymph nodes located behind the manubrium (subsequently called level VII) were included by the American Joint Committee as part of the central compartment of the neck, despite their anatomic location at the superior mediastinum [25].

As the CNC includes both level VI and VII, complete CNC dissection involves an area covering the hyoid bone superiorly, the brachiocephalic vessels caudally, and the carotid sheaths laterally [19] (Figure 1).

## 3. Rate of CNC Metastases in Laryngeal and Hypopharyngeal SCCs

The CNC is not encompassed in radical, modified radical, or selective dissections of the neck [7]. A relevant number of studies assessed the frequency and pattern of lymph node metastasis in the lateral neck, whereas the prevalence of the involvement of the anterior compartment of the neck in laryngeal and hypopharyngeal tumors, is not fully defined. As a result, up-to-date, universally accepted guidelines for the elective surgical management of CNC in non-thyroid malignancies are lacking, and the available data about prevalence and prognostic implications related to CNC involvement are invariably limited and biased.

In particular, when CNC was referred to as ‘PTLN’, the description of the inferior limit was often missing. Furthermore, the selection criteria for the CNC dissection were rarely defined (many reported as “according to the surgeon preference”), and CNC dissection has not been routinely performed in most of the analyzed series. Consequently, comparisons among institutions’ results are invariably difficult, and this is highlighted by the high variability in the prevalence of PTLN metastasis reported by different series (Table 1). Overall, metastasis in CNC lymph nodes in laryngeal and hypopharyngeal SCCs were reported to occur from 10 to 30% of patients, variably differing according to the site, subsite, and stage of the primary tumor [11,23,26].

### 3.1. Site

The overall PTLN metastasis rate reported for laryngeal cancers ranges from 13.5% [27] to 20% [11,26]. For hypopharyngeal cancers, the rate is more variable, ranging from 8% [23] to 38% [28]. In the published literature, the prevalence of PTLN metastasis results are higher for hypopharyngeal compared with laryngeal cancers [11,26,28], even though statistical significance was not detected. A recent systematic review [29] confirmed this trend; however, the statistical difference comparing the pooled data among laryngeal and hypopharyngeal carcinomas was still not significant (*p* = 0.157).

### 3.2. Subsite

Many authors investigated the possible correlation between subsite location and PTLN involvement. In laryngeal cancers, the subglottic extension has been linked to PTLN disease by many authors [9,11,23,26,30]; a recent meta-analysis proved on univariate analysis that subglottic tumors have a significantly higher rate of PTLN metastasis than tumors confined in glottis and supraglottis.

Hypopharyngeal cancer subsites have been studied for their possible predictive role in the central neck involvement. Notably, the literature shows highly variable results; piriform sinus invasion, with special reference to its apex [2,27,31,32], showed a correlation to PTLN metastasis in different series [33], with proven statistical significance on multivariate analysis [31]. However, in the cohort of Joo et al. [2], post-cricoid tumors displayed a significantly higher rate of PTLN disease compared with piriform sinus malignancies. Notably, relevant drawbacks have been pointed out in this study, such as the low number of PTLN metastasis in the series, and the inequal distribution between subsites of hypopharyngeal tumors.

The posterior hypopharyngeal wall was found to be the subsite at highest risk for PTLN disease [33], but statistical analysis did not reach significance.

### 3.3. Occult Metastasis

The prevalence of occult metastasis has been investigated by few authors. Studying hypopharyngeal malignancies, Takooda et al. [34] found an occult metastasis rate of 41.7%. Over the years, a decreasing rate has been reported [32], and more recent studies showed an occult node metastasis rate of 12% [27] −14% [31,35].

A recent meta-analysis reported an overall rate of occult nodal metastasis of 9.2%, resulting from the analysis of the only two studies in which elective PTLN dissection was performed along with salvage total laryngectomy for laryngeal SCCs [29].

It is reasonable to assume this high variability can be explained in terms of imaging evolution, considering that the first study mentioned above was published in the 1990s.

### 3.4. T Category and Characteristics

With respect to the T category of the primary tumor, there is statistically significant evidence showing that advanced T is a risk factor for CNC tumor spread. As reported in Chabrillac et al. meta-analysis [29], T4 tumors were more prone to spread to PTLN compared with T3 and T2 malignancies. Similarly, Tomoika et al. [33] confirmed a significant trend to PTLN metastasis for advanced cT hypopharyngeal cancers. Even though not statistically significant, Dequanter et al. [28] reported that PTLN positivity was invariably detected in hypopharyngeal cancers exceeding 35 mm. Notably, many studies presented in this review were focused on advanced T category tumors; thus, one can argue that little is known about CNC involvement in early-stage laryngeal and hypopharyngeal cancers.

For laryngeal cT4N0 tumors, it seems reasonable to include elective CNC dissection when the gross extralaryngeal component piercing the cartilage extends below the axial plane passing at the upper edge of the cricoid cartilage, and/or subglottic intralaryngeal extension is found, and/or the tumor invades the hypopharynx [29] (Figure 2).

### 3.5. Bilateral Involvement

Bilateral involvement of the CNC has been poorly investigated. Notably, many authors omitted the dissection of CNC contralateral to the primary tumor. Consequently, the scarce and insufficient data available further hamper the effort to detect significant risk factors for contralateral PTLN disease and thus establish robust recommendations.

Bilateral/contralateral metastasis were found to be relatively rare [2,9,31,35,36]. Moreover, author indications for bilateral CNC dissection remarkably differed in the literature. Some groups suggested PTLN dissection for hypopharyngeal malignancies with advanced cT [33] or cN categories [37], particularly if involving the post-cricoid area, posterior hypopharyngeal wall area, or cervical oesophagus [33]. Moreover, the putative advantages of bilateral PTLN should be considered together with the risk of pharyngo-cutaneous fistula, especially during salvage surgery [36].

### 3.6. Correlation between Lateral and Central Neck Involvment

Association between the lateral and central neck involvement has been more widely explored. Lateral neck metastases (levels I-V) were found to be a significant predictor for PTLN involvement both in laryngeal [9,35,38] and hypopharyngeal cancers [2,26,33,37]. These results emphasize the importance of CNC dissection in a clinically positive neck, regardless of the primary tumor location.

However, few studies also showed that PTLN involvement can occur as first echelon, independently from lateral neck involvement [11,35]. Timon et al. [11] showed high risk of PTLN disease in node-negative lateral neck in post-cricoid hypopharyngeal and cervical oesophagus carcinomas, but statistical significance was not reached. Importantly, both series [11,35] included previously irradiated patients, and one could argue that the primary treatment had altered the usual lymphatic routes of tumor spread.

**Table 1 cancers-15-00804-t001:** Rate of central neck compartment involvement (author, year, site and subsite of the primary, category of the primary and treatment setting, overall central neck metastasis rate, occult central neck metastasis rate). CL, contralateral; IL, ipsilateral; NR, not reported; PTLN, paratracheal lymph nodes.

First Author (Year)	N. of Cases	Primary Site of Tumor (n)	Lymph Node Chain	Metastases (%)	Level VI Metastases by Primary Tumor Site (%)	TNM	Postoperative Recurrence in Patients with Level VI N+
Takooda (1992)	70	Piriform sinus: 46 Postcricoid: 24	PLTN	41.7%	Postcricoid: 41.7% Piriform sinus: NR	T1N0M0-T4N3M0	NR
Weber (1993)	141	Cervical oesophagus: 14 Larynx: 91 Hypopharynx: 36	PTLN	20.5%	Cervical oesophagus: 71.4% Larynx: 17.6% Hypopharynx: 8.3%	T1N0M0-T4N3M0	48%
Shenoy (1994)	45	Glottic	PTLN	IL: 9% CL: 4.5%	-	T3/T4	NR
Yang (1998)	21	Glottic	PTLN	4.7%	-	T3/T4	NR
Timon (2003)	50	Larynx: 20 Postcricoid/cervical oesophagus: 21 Pyriform fossa/lateral pharyngeal wall: 9	PLTN	26%	Larynx: 20% Postcricoid/cervical oesophageal region: 43% Pyriform fossa/lateral pharyngeal wall: 0%	T4	Peristomal: 23%
Petrovic (2004)	174	Laryngeal	PTLN	9%	NR	T3/T4	Peristomal: 15%
Plaat (2005)	85	Hypopharyngeal/cervical oesophagus: 20 Subglottic: 3 Supraglottic: 20 Glottic: 42	PTLN	24%	Hypopharyngeal/cervical oesophageal: 35% Subglottic: 67% Supraglottic: 30% Glottic: 12% Larynx w/subglottic extension: 27%	T3/T4 N0-N3	Peristomal: 15%
Guo (2005)	108	Hypopharynx Pyriform sinus: 100	PTLN	8.6%	NR	T1N0M0-T4N3M0	NR
Garas (2006)	15	Subglottic	PTLN	26.6%	-	T3/T4	NR
Joo (2009)	64	Postcricoid: 7 Pyriform sinus: 45 Posterior pharyngeal wall: 11	PTLN	IL: 14/64 (22%) CL: 1/42 (2%)	Postcricoid: 57% Pyriform sinus: 20% Posterior pharyngeal wall: 8%	T1N0M0-T4N3M0	NR
Peters (2012)	149	Supraglottic: 45 Glottic: 61 Subglottic: 9 Hypopharyngeal: 30 Cervical oesophagus: 3 Neopharyngeal: 1	PTLN	15%	Supraglottic: 9% Glottic: 20% Subglottic: 11% Hypopharyngeal/cervical oesophagus: 16%	T1N0M0-T4N3M0	NR
Dequanter (2013)	31	Laryngeal: 18 Hypopharyngeal: 13	PTLN	IL: 19% CL: 0%	Laryngeal: 0% Hypopharyngeal: 46%	T3/T4	NR
Eun-Jae Chung (2014)	39	Medial wall pyriform sinus	PTLN	16.2%	-	T1N0M0-T4N3M0	NR
Resta (1985)	124	Pyriform sinus: 37 Transglottic: 20 Glottic w/subglottic extension: 8 Glottic: 11 Supraglottic: 48	Delphian node	21%	Pyriform sinus: 35% Transglottic: 30% Glottic w/subglottic extension: 38% Glottic: 18% Supraglottic: 4%	NR	81%
Olsen (1987)	20	Laryngeal	Delphian node	20/20 (100%)	Glottic: 13/20 (65%) Supraglottic: 2/20 (10%) Transglottic: 1/20 (5%) Subglottic: 4/20 (20%)	T1N0M0-T3N2M0	70%
Szmeja (1995)	109	Laryngeal	Delphian node	7%	NR	NR	NR
Thaler (1997)	92	Larynx	Delphian node	8.7%	NR	T1N0M0-T4N3M0	Stomal: 38%
Tomik (2001)	1400	Laryngeal	Delphian node	1.7%	NR	T1N0M0-T4N3M0	NR
Murono (2009)	1	Laryngeal	Delphian node	1/1 (100%)	-	NR	NR
Nakayama (2011)	65	Larynx	Delphian node	4.6%	NR	pT3N2b, pT4N2c, T3N1	33%
Buckley (2000)	100	Supraglottic: 33 Glottic: 22 Multiregional: 20 Hypopharyngeal: 25	NR	N0: IL: 7/34 (21%) CL: 3/30 (10%) N+: IL: 4/15 (27%) CL: 2/11 (18%)	NR	T1N0M0-T4N3M0; N0: 58 N+:42	NR
Shen (2007)	102	Pyriform sinus	NR	1.3%	-	T1N0M0-T4N3M0	NR
Lou (2008)	106	Hypopharynx	NR	5.7%	NR	T1N0M0-T4N3M0	NR

## 4. Prognostic Role of CNC Metastasis

Many authors explored the prognostic role of the CNC involvement in terms of overall survival (OS), disease-specific survival (DSS), and recurrence-free survival (RFS).

Most of the studies demonstrated that the CNC positivity for metastatic disease adversely affected the oncological outcomes. Nevertheless, statistical significance was not always reached [2,11,37] and the effective prognostic impact of the CNC involvement in laryngeal and hypopharyngeal cancers remains unclear.

In terms of OS, CNC involvement resulted in a significant adverse prognostic factor in various series [23,31,39], showing a five-year OS of 31.6% in patients who had CNC metastasis (vs. 73.5% in those without CNC involvement) [31]. Some other authors showed that the presence of extranodal extension (ENE) in PTLN was the most significant prognostic factor for OS [26,40].

In relation to DSS, Chung et al. [31] showed a significantly lower five-year DSS rate in patients with CNC involvement than in those with uninvolved CNC (26.3% vs. 55.1%). Similarly, but without reaching statistical significance, Joo et al. [2] showed a five-year DSS of 29% in patients with PTLN involvement, compared with 60% in patients without PTLN disease. Another significant result was achieved in a multivariate analysis that showed that elective PTN dissection was associated with improved OS, DSS, and RFS in the setting of salvage surgery [35]. However, a significant bias must be considered, as PTLN dissection was performed according to ‘the surgeon discretion’.

CNC involvement also demonstrated a role in terms of recurrence, particularly peristomal. Peristomal recurrence following total laryngectomy is a fearful complication and it is usually associated with a fatal outcome. Several factors are known to predispose to peristomal recurrence (e.g., preoperative tracheostomy, previous conservative treatment, subglottic tumor extension), including PTLN metastasis [23].

Patients with a history of tracheostomy before laryngectomy have a poor prognosis; the stoma may be seeded with tumor cells and therefore it should be included within the resection together with a bilateral PTLN dissection [23].

The overall rate of peristomal recurrences highlighted in a recent meta-analysis [29] was 3.4%. Several studies demonstrated increased peristomal recurrence in patients who had PTLN involvement. In the Timon et al. series [11], all peristomal recurrence were in patients with previously detected PTLN disease, which was frequently associated with ENE. Dogan et al. showed a significant association between PTLN involvement and peristomal recurrence [41].

Notably, the use of radiotherapy has been suggested to be a stronger treatment than PTLN dissection [29], but no evidence supported this hypothesis yet [33]. In our view, adjuvant C(RT) should effectively control microscopic lymphatic disease at CNC. However, given that gross millimeter/infracentimeter metastases are not detected on preoperative imaging, there will be at least 4 weeks between surgery and adjuvant treatment, and that full doses of RT will most likely not be delivered to the full CNC region in the absence of pathological information, we believe that elective CNC dissection followed (when needed) by adjuvant (C)RT should offer more than adjuvant (C)RT alone, for patients at high risk for CNC occult lymphatic metastases. Moreover, in various studies, a lower (but not statistically significant) peristomal recurrence rate was observed in patients who underwent PTLN dissection [9,30,39].

## 5. Radiological Assessment of CNC

The majority of PTLN metastases are usually less than 1 cm in diameter; thus, preoperative detection of pathological nodal disease is challenging both clinically and radiologically. For this reason, in a series published prior to the 1980s, preoperative detection of pathological PTLN metastasis was exceedingly difficult.

Even nowadays, although imaging of levels I–V has been studied extensively [42,43], data regarding CNC imaging are scarce, and the existing evidence attested that the radiological assessment does not provide a satisfactory accuracy, due to the high false-negative rates [31].

Another challenge is represented by the uncertainty that the scoring criteria used for the lateral neck can be applied in the CNC. In this regard, attempts to identify CT and MRI criteria to predict PTLN metastasis have been made [44]. Peters et al. investigated a possible cut off in minimal and maximal axial diameters, attesting a maximal axial diameter of 5 mm or more as the best predictor of PTLN and showing in CT and MRI a sensitivity of 70% and 50% and a specificity of 36% and 71%, respectively.

Sensitivity, specificity, positive/negative predictive values, and accuracy of CT, MRI, US, and PET in the detection of CNC metastasis have been investigated by Park et al. [27]. CT combined with US demonstrated the best results in terms of sensitivity (51.7%) and negative predictive value (88%). US-guided fine needle aspiration biopsy (FNAB) may add useful information and further increase the sensitivity of ultrasonography itself [45].

On the other hand, adding MRI to CT did not result in improvement of the diagnostic accuracy and PET showed unfavorable results compared with all the other imaging techniques. PET/CT was also confirmed to be unreliable in detecting metastases in the CNC, with a low sensitivity (58%) [38] not significantly different from the sensitivity provided by CT/MRI techniques (42%).

Currently, one can conclude that intraoperative exploration and pathological examination seem to represent the best ways to estimate the state of PTLN.

## 6. Complications Related to Central Neck Compartment Dissection

Complications related to CNC dissection are considered, by many authors, as the main argument against elective treatment of this region [29]. Complications usually related to CNC dissection are pharyngo-cutaneous fistula, hypothyroidism, hypoparathyroidism, and, in organ preservation surgery, laryngeal nerves damage. Moreover, CNC dissection requires increased operative time, which is nowadays largely accepted as a predictor of complications in head and neck surgery [46].

In the series of Van der Putten et al. [36], 58% of patients treated with salvage surgery and neck dissection developed complications, the majority being pharyngo-cutaneous fistulas. The authors observed a significant increase in complications in patients treated with bilateral PTLN dissection, along with salvage surgery, whereas unilateral CCN dissection did not result in being associated with a higher risk of complications. In contrast with these findings, Lucioni et al. [9] reported no significantly increased rate of complications in patients receiving bilateral PTLN dissection. Nevertheless, the populations studied in each series were different, respectively including 95% patients undergoing salvage surgery vs. 27%. This could represent a reasonable explanation for the discrepancy.

Hypoparathyroidism is a well-studied complication in CNC dissection. A recent study [47] showed a significant increase of long-term hypoparathyroidism following total laryngectomy. However, this observation was not confirmed by other authors [48]. Similarly, hypothyroidism was observed to be significantly higher in CNC dissection [49], but it has not been confirmed by other studies [48,50].

Primary vs. salvage treatment setting is a fundamental factor for the risk of complications, and it should be balanced for elective CNC dissection indication. As incidence of lymph node metastases observed in the literature in the contralateral CCN side is low, patients treated in a salvage setting should be carefully selected for bilateral PTLD dissection [36].

Finally, the surgeon should be aware that extensive CNC dissection, especially when in bilateral and post (chemo)radiotherapy, may impair the blood supply of the cranial trachea, with possible tracheal necrosis after total laryngectomy or total pharyngolaryngectomy. Even if this complication is not extensively reported in the literature, we warrant to carefully preserve the outer tracheal perichondrium avoiding tracheal peeling with hot instruments. Furthermore, intraoperative viability of the trachea should be assessed before completing the permanent tracheostomy, and drains preventing paratracheal fluid collection should always be in place.

## 7. Conclusions

The most relevant unanswered question is when and how to electively dissect the CNC, given that the reliability of imaging systems remains suboptimal.

To delineate rational indications for the CNC dissection, knowledge of the most significative risk factors for CNC metastasis remains indispensable. Primary carcinoma of hypopharynx with involvement of pyriform sinus apex, retro-cricoid area, or cervical esophagus, T4 category, node-positive lateral neck, and subglottic involvement are statistically significant risk factors for PTLN disease [29].

In laryngeal cancers, elective CNC dissection should be considered for advanced T tumors, node-positive lateral neck, gross extralaryngeal extension below the axial plane passing at the superior border of the cricoid cartilage, and tumor arising from or extending to the subglottic region (Figure 3). With the awareness of the poor existing data regarding laterality of central neck involvement, bilateral elective CNC dissection could be considered in tumors involving the posterior hypopharyngeal wall, post-cricoid area, cervical oesophagus, and/or midline larynx. If a bilateral CNC dissection is indicated, it is also recommended to consider the non-negligible risks of complications, particularly for salvage surgery. Furthermore, as peristomal recurrence after total laryngectomy is highly related to a fatal outcome, a complete PTNL dissection in a patient who received tracheotomy before definitive surgical treatment seems a reasonable strategy.

## Figures and Tables

**Figure 1 cancers-15-00804-f001:**
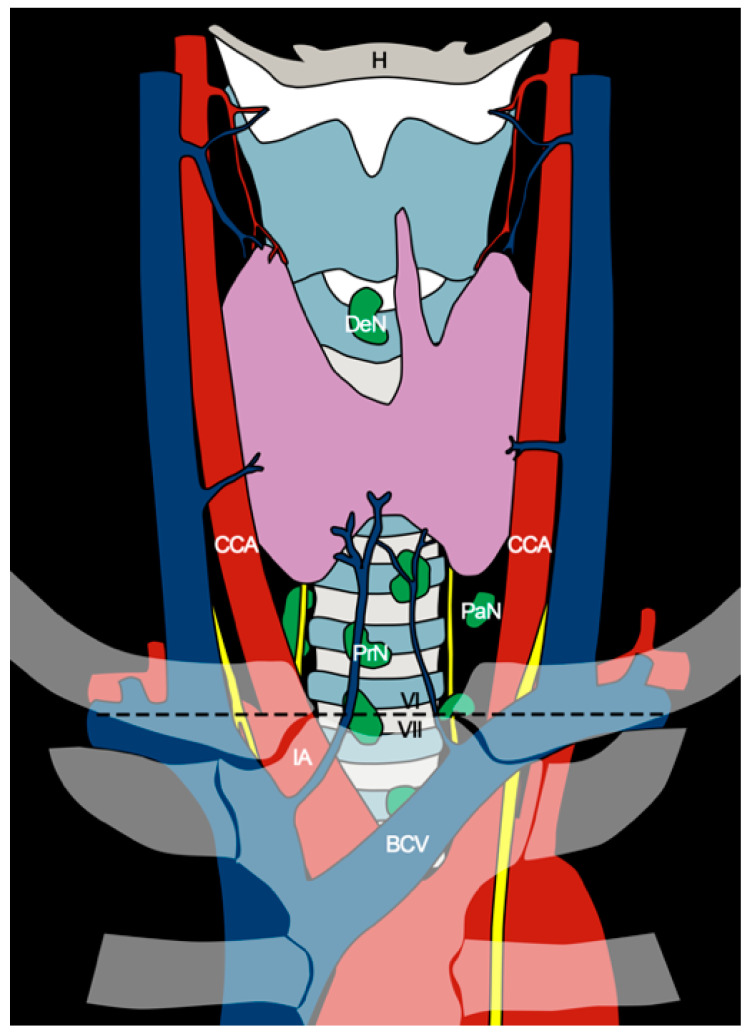
Anatomical diagram of the central neck compartment. The axial plane passing through the sternal notch (black dashed line) separates levels VI and VII. CCA, common carotid artery; DeN, Delphian node; H, hyoid bone; IA, innominate artery; BCV, brachiocephalic vein; PaN, paratracheal nodes; PrN, pretracheal nodes.

**Figure 2 cancers-15-00804-f002:**
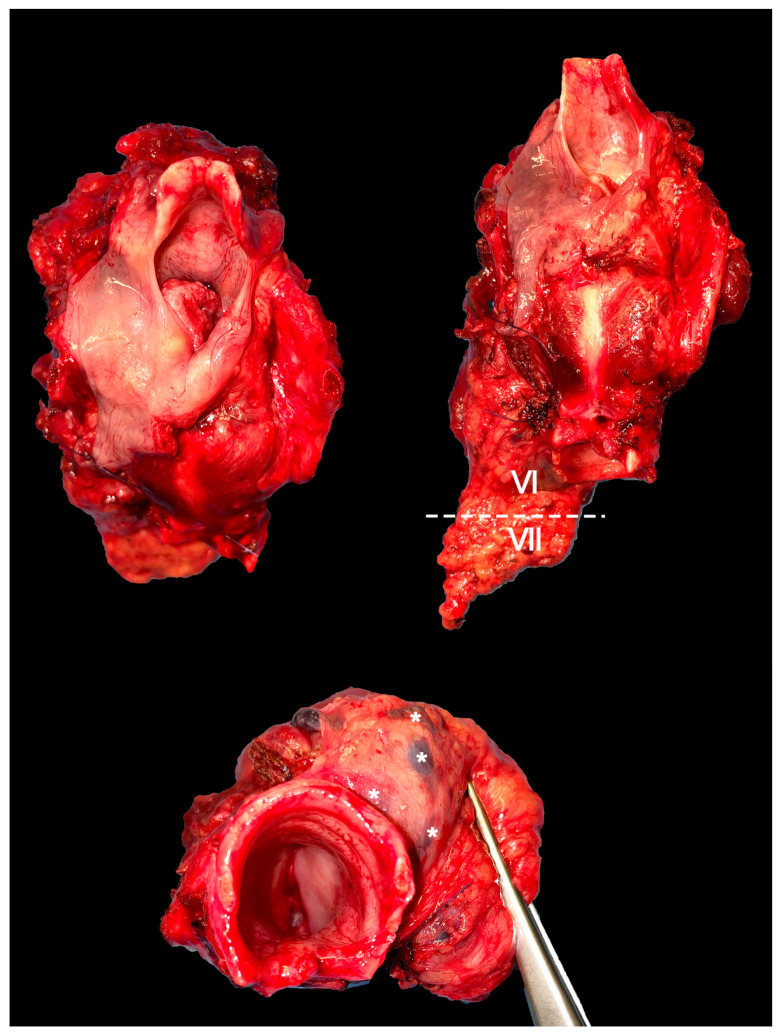
Panel showing a total laryngectomy specimen with en bloc dissection of the VI and VII level ipsilateral to the tumor. Several lymph nodes (*) can be observed in the paratracheal tissue.

**Figure 3 cancers-15-00804-f003:**
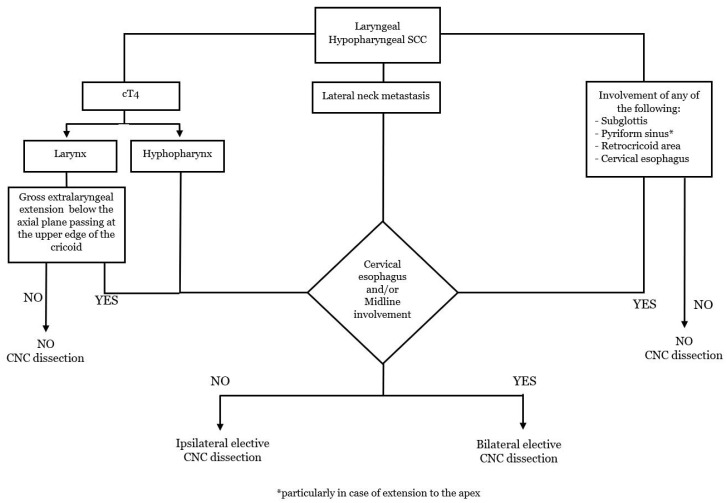
Flow chart: indications for the elective dissection of the CNC.

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
