# Peer review of "Central Compartment Neck Dissection in Laryngeal and Hypopharyngeal Squamous Cell Carcinoma: Clinical Considerations"

_cancers, 2023, doi:10.3390/cancers15030804_

Round 1

Reviewer 1 Report

This is a detailed and extensive review of undiscussed paratracheal dissection of laryngeal and hypopharyngeal SqCC, due to lack of sufficient data. I think it will be extremely useful for the readers, the head and neck surgeons.

I'll point out some of the things I noticed.

P5-L166

Reference numbers are still missing.

P6-L204, 206, 209, 219

The description of “T stage” or “N stage” is misuse.

The AJCC cancer staging manual (eight edition, page 6) states ‥‥the term stage should not be used to describe individual T, N, or M category designations that often are mistakenly referred to as "stage".

P9-L308 or later

There is no mention of the possibility that extensive CNC dissection, especially bilateral dissection, may increase the incidence of tracheal necrosis after total laryngectomy or total pharyngolaryngectomy. It may give the readers useful information.

P10-L334

All others are written as “PTLN dissection”, but only here is written as “PTND dissection”.

P11-L365

Patients with a history of tracheostomy have a poor prognosis, and bilateral PTLN dissection is a reasonable strategy, which is clinically very useful. We believe that the rationale for this important recommendation should be presented in this review.

Author Response

Reviewer 1

We thank the reviewer for the time and efforts spent revising our manuscript and for the constructive and pertinent comments. We modified the paper according to the suggested revisions.

P5-L166

Reference numbers are still missing.

Missing reference numbers were added

P6-L204, 206, 209, 219

The description of “T stage” or “N stage” is misuse.

The AJCC cancer staging manual (eight edition, page 6) states ‥‥the term stage should not be used to describe individual T, N, or M category designations that often are mistakenly referred to as "stage".

Indeed, the text was corrected

P9-L308 or later

There is no mention of the possibility that extensive CNC dissection, especially bilateral dissection, may increase the incidence of tracheal necrosis after total laryngectomy or total pharyngolaryngectomy. It may give the readers useful information.

Yes indeed, thank you for this remark. We added the following considerations in the session “Complications related to central neck compartment dissection”:

“Finally, the surgeon should be aware that extensive CNC dissection, especially when bilateral and post (chemo)radiotherapy, may impair the blood supply of the cranial trachea, with possible tracheal necrosis after total laryngectomy or total pharyngolaryngectomy. Even if this complication is not extensively reported in the literature, we warrant to carefully preserve the outer tracheal perichondrium avoiding tracheal peeling with hot instruments. Furthermore, intraoperative viability of the trachea should be assessed before completing the permanent tracheostomy, and drains preventing paratracheal fluid collection should always be placed.”

P10-L334

All others are written as “PTLN dissection”, but only here is written as “PTND dissection”.

Corrected

P11-L365

Patients with a history of tracheostomy have a poor prognosis, and bilateral PTLN dissection is a reasonable strategy, which is clinically very useful. We believe that the rationale for this important recommendation should be presented in this review.

Yes indeed. At the end of the session “Conclusions” we briefly stated that …”as peristomal recurrence after total laryngectomy is highly related to a fatal outcome, a complete PTNL dissection in patient who received tracheotomy before definitive surgical treatment seems a reasonable strategy”.

In order to implement the rationale of this recommendation, we added the following sentence with citation in the session “T category and characteristics”

“Patients with a history of tracheostomy before laryngectomy have a poor prognosis, the stoma may be seeded with tumor cells and therefore it should be included within the resection together with a bilateral PTLN dissection [23].”

Reviewer 2 Report

This is a weel written and thorough work. The topic is interesting, debated and have clinical implications. The paper is well organized in different sections. Limitations of the study are outlined and conclusions with possible clinical indications are clear.

I have only two considerations:

- methodology: this is a review but I'd like that the authors would clarify in the text how the review was conducted

- subsite: authors try to identify the different rate of central node metastasis depending on subsite tumor involvement. They don't report about anterior extension with cartilage invasion. This is reported as an historical risk factor for prelaryngeal and pretracheal node metastasis and some authors advocate use of thyroidectomy in order to allow a comprehensive dissection of this area. I'd like a comment by them about this issue and, if they deem it appropriate, include it in the paper.

Author Response

Reviewer 2

We thank the reviewer for the time and efforts spent revising our manuscript and for the constructive and pertinent comments. We modified the paper according to the suggested revisions.

This is a weel written and thorough work. The topic is interesting, debated and have clinical implications. The paper is well organized in different sections. Limitations of the study are outlined and conclusions with possible clinical indications are clear.

I have only two considerations:

- methodology: this is a review but I'd like that the authors would clarify in the text how the review was conducted

Yes, we clarified in the text that this review was narrative.

- subsite: authors try to identify the different rate of central node metastasis depending on subsite tumor involvement. They don't report about anterior extension with cartilage invasion. This is reported as an historical risk factor for prelaryngeal and pretracheal node metastasis and some authors advocate use of thyroidectomy in order to allow a comprehensive dissection of this area. I'd like a comment by them about this issue and, if they deem it appropriate, include it in the paper.

Thank you for this comment. We revised the flowchart and we added the following considerations in the session T category and characteristics:

“For laryngeal cT4N0 tumors, it seems reasonable to include elective CNC dissection when the gross extralaryngeal component piercing the cartilage extends below the axial plane passing at the upper edge of the cricoid cartilage, and/or subglottic intralaryngeal extension is found, and/or the tumor invades the hypopharynx [29] (Figure 2).”

Reviewer 3 Report

Entitled "Central compartment neck dissection in laryngeal and hypo-pharyngeal squamous cell carcinoma: clinical considerations", this paper describes the current knowledge about the management of central compartment lymph nodes in the context of head and neck cancer and is a very complete and interesting review of the literature on this topic.

I really appreciate the author's work and it is a well written and organised work. Although the data provided is not new and has already been described in the literature, the authors have tried to synthesise the knowledge on the subject and may help in tumour board decisions.

I suggest some changes:

- The discussion should be shorter. The authors repeat data that have already been reported previously and some that can be included as an introduction. Rather than a conclusion it seems like a summary of the article as it stands.

- The flow diagram in figure 2 should be clarified. In the current mode it appears that, for example, in a T4 laryngeal tumour the central compartment should be treated only if there is invasion of the cervical oesophagus or the midline. It should be improved.

- Figure 1 could be accompanied by a diagram clearly marking the anatomical boundaries of the areas. This would allow the text of the article to be reduced.

- On line 166, a reference is missing

- The bibliography should be adapted to the standards of the journal.

Author Response

Reviewer 3

We thank the reviewer for the time and efforts spent revising our manuscript and for the constructive and pertinent comments. We modified the paper according to the suggested revisions.

Entitled "Central compartment neck dissection in laryngeal and hypo-pharyngeal squamous cell carcinoma: clinical considerations", this paper describes the current knowledge about the management of central compartment lymph nodes in the context of head and neck cancer and is a very complete and interesting review of the literature on this topic.

Thank you for this positive comment

I really appreciate the author's work and it is a well written and organised work. Although the data provided is not new and has already been described in the literature, the authors have tried to synthesise the knowledge on the subject and may help in tumour board decisions.

I suggest some changes:

- The discussion should be shorter. The authors repeat data that have already been reported previously and some that can be included as an introduction. Rather than a conclusion it seems like a summary of the article as it stands.

The discussion has been shortened and repetitive parts were erased.

- The flow diagram in figure 2 should be clarified. In the current mode it appears that, for example, in a T4 laryngeal tumour the central compartment should be treated only if there is invasion of the cervical oesophagus or the midline. It should be improved.

Yes, it seems reasonable to omit elective CNC dissection when the extralaryngeal extension stays above the axial plane passing at the upper edge of the cricoid cartilage and no subglottic intralaryngeal extension is found. We improved the diagram, we also added the following comment in the session “T category and characteristics”:

“For laryngeal cT4N0 tumors, it seems reasonable to include elective CNC dissection when gross extralaryngeal component extends below the axial plane passing at the upper edge of the cricoid cartilage, and/or subglottic intralaryngeal extension is found, and/or the tumor invades the hypopharynx [29] (Figure 2).”

- Figure 1 could be accompanied by a diagram clearly marking the anatomical boundaries of the areas. This would allow the text of the article to be reduced.

We added a new diagram marking the anatomical boundaries, and we reduced the text

- On line 166, a reference is missing

Reference n.29 was added

- The bibliography should be adapted to the standards of the journal.

The reference list was revised and adapted to the standards of the journal

Reviewer 4 Report

This narrative review evaluates the need or not for central neck compartment dissection in laryngeal and hypopharyngeal carcinomas. It is a debated topic because there is not enough evidence to support the indications for its practice, so the work is relevant.

The work is well structured, correctly written and documented. The main problem I see is that it does not contribute anything new with respect to a recent systematic review and meta-analysis on the same subject published in Oral Oncology: Chabrillac, E.; Jackson, R.; Mattei, P.; D'Andréa, G.; Vergez, S.; Dupret-Bories, A.; Edafe, O. Paratracheal Lymph Node Dissection during Total (Pharyngo-)Laryngectomy: A Systematic Review and Meta-Analysis. Oral Oncol. 2022, 132, 106017, doi:10.1016/j.oraloncology.2022.106017 (ref 30 of the paper). 

Minor aspects:

- On page 5, line 166, the reference is missing; I imagine they mean number 30.

- Given that almost all cases in which elective dissection of the central compartment would be indicated will require adjuvant (C)RT, one aspect that should be discussed is the possibility of elective treatment of this compartment by (C)RT, which would avoid the possible complications associated with this procedure.  

Author Response

Reviewer 4

We thank the reviewer for the time and efforts spent revising our manuscript and for the constructive and pertinent comments. We modified the paper according to the suggested revisions.

This narrative review evaluates the need or not for central neck compartment dissection in laryngeal and hypopharyngeal carcinomas. It is a debated topic because there is not enough evidence to support the indications for its practice, so the work is relevant.

Thank you for this nice comment

The work is well structured, correctly written and documented. The main problem I see is that it does not contribute anything new with respect to a recent systematic review and meta-analysis on the same subject published in Oral Oncology: Chabrillac, E.; Jackson, R.; Mattei, P.; D'Andréa, G.; Vergez, S.; Dupret-Bories, A.; Edafe, O. Paratracheal Lymph Node Dissection during Total (Pharyngo-)Laryngectomy: A Systematic Review and Meta-Analysis. Oral Oncol. 2022, 132, 106017, doi:10.1016/j.oraloncology.2022.106017 (ref 30 of the paper). 

Although the data here provided are not new and were already reported in the literature, we tried to synthesize the current knowledge on the subject, to support the decision making processes in clinical practice.

Minor aspects:

- On page 5, line 166, the reference is missing; I imagine they mean number 30.

Missing reference was added

- Given that almost all cases in which elective dissection of the central compartment would be indicated will require adjuvant (C)RT, one aspect that should be discussed is the possibility of elective treatment of this compartment by (C)RT, which would avoid the possible complications associated with this procedure. 

The answer to this opinion cannot be definitive, since no solid data are available. We revised the paragraph “Prognostic role of CNC metastasis” that addressed this aspect.

“Notably, the use of radiotherapy has been suggested to be a stronger treatment than PTLN dissection [29], but no evidence supported this hypothesis yet [33].  In our view, adjuvant C(RT) should effectively control microscopic lymphatic disease at CNC. However, given that gross millimeter/infracentimeter metastases are not detected on preoperative imaging, that there will be at least 4 weeks between surgery and adjuvant treatment, and that full doses of RT will most likely not be delivered to the full CNC region in the absence of pathological information, we believe that elective CNC dissection followed (when needed) by adjuvant (C)RT should offer more than adjuvant (C)RT alone, for patients at high risk for CNC occult lymphatic metastases. Moreover, in various studies, a lower (but not statistically significant) peristomal recurrence rate was observed in patients who underwent PTLN dissection [9,30,39].”